

# The role of 2,4-dihydroxyquinoline (DHQ) in *Pseudomonas aeruginosa* pathogenicity

Jordon D. Gruber[1], Wei Chen[1], Stuart Parnham[1], Kevin Beauchesne[2], Peter Moeller[2], Patrick A. Flume[3] and Yong-Mei Zhang[1]

[1] Department of Biochemistry and Molecular Biology, Medical University of South Carolina, Charleston, SC, United States
[2] Natural Products Chemistry, National Ocean Service, Charleston, SC, United States
[3] Department of Medicine, Medical University of South Carolina, Charleston, SC, United States

## ABSTRACT

Bacteria synchronize group behaviors using quorum sensing, which is advantageous during an infection to thwart immune cell attack and resist deleterious changes in the environment. In *Pseudomonas aeruginosa*, the *Pseudomonas* quinolone signal (Pqs) quorum-sensing system is an important component of an interconnected intercellular communication network. Two alkylquinolones, 2-heptyl-4-quinolone (HHQ) and 2-heptyl-3-hydroxy-4-quinolone (PQS), activate transcriptional regulator PqsR to promote the production of quinolone signals and virulence factors. Our work focused on the most abundant quinolone produced from the Pqs system, 2,4-dihydroxyquinoline (DHQ), which was shown previously to sustain pyocyanin production and antifungal activity of *P. aeruginosa*. However, little is known about how DHQ affects *P. aeruginosa* pathogenicity. Using *C. elegans* as a model for *P. aeruginosa* infection, we found *pqs* mutants only able to produce DHQ maintained virulence towards the nematodes similar to wild-type. In addition, DHQ-only producing mutants displayed increased colonization of *C. elegans* and virulence factor production compared to a quinolone-null strain. DHQ also bound to PqsR and activated the transcription of *pqs* operon. More importantly, high extracellular concentration of DHQ was maintained in both aerobic and anaerobic growth. High levels of DHQ were also detected in the sputum samples of cystic fibrosis patients. Taken together, our findings suggest DHQ may play an important role in sustaining *P. aeruginosa* pathogenicity under oxygen-limiting conditions.

# INTRODUCTION

*Pseudomonas aeruginosa* possesses a versatile metabolic system, which facilitates fitness of the organism to survive in diverse environments. People frequently come in contact with *P. aeruginosa*, but an intact immune system can block the bacterium from establishing an infection. However, both immunocompetent and immunocompromised individuals

Corresponding author
Yong-Mei Zhang,
zhangym@musc.edu

are still at risk of *P. aeruginosa* infections, particularly those with diabetes, burn wounds, an implanted medical device, or use contaminated contact lenses (*Chatterjee et al., 2014*; *Gaynes & Edwards, 2005*; *Gordon et al., 2001*). Patients with the genetic disorder cystic fibrosis (CF) are especially susceptible to *P. aeruginosa* lung infection, which is correlated with increased morbidity and mortality (*Hoiby, 2011*). In the lungs of a CF patient, *P. aeruginosa* replicates to high densities and forms antibiotic-resistant biofilms, which is a community lifestyle protected by exopolymeric substance (*Palmer et al., 2005*). Treatment of chronic and acute *P. aeruginosa* infections is difficult due to various intrinsic antimicrobial resistance mechanisms, including cell envelope barrier, efflux pumps, and high mutation rate (*Henrichfreise et al., 2007*; *Mandsberg et al., 2009*). Compared with classical antibiotics that target essential cellular processes for survival, an alternative approach is to target cellular systems that are responsible for virulence. This strategy may allow the host's immune system to naturally combat the infection, while not imposing selective pressure for resistant mutants.

Quorum sensing (QS) is a form of bacterial intercellular communication that synchronizes group behaviors and is essential for *P. aeruginosa* to establish an infection (*Rumbaugh et al., 1999*). QS molecules bind to their cognate transcriptional regulators at a critical intracellular concentration to promote virulence factor production. The Las and Rhl systems, commonly found in gram-negative bacteria, produce acylhomoserine lactones (AHLs) that activate LuxR-type transcriptional regulators LasR and RhlR (*Fuqua & Greenberg, 2002*). The Pqs system, which is limited to *P. aeruginosa* and other closely related bacteria, produces alkylquinolones that activate a LysR-type transcriptional regulator PqsR in *P. aeruginosa* (*Diggle et al., 2006*). Among the greater than 50 different quinolones produced by the Pqs system, only alkylquinolones HHQ and PQS have established functions for signaling and promoting pathogenicity, such as activating the production of pyocyanin and biofilm formation (*Deziel et al., 2004*).

Alkylquinolones are synthesized by enzymes encoded by the *pqs* operon and other *pqs* genes located elsewhere on the genome. PqsA catalyzes the first step in quinolone synthesis by converting anthranilic acid, an intermediate of tryptophan biosynthesis, to anthraniloyl-CoA (*Coleman et al., 2008*). The next step involves the transfer of the anthraniloyl-moiety to PqsD and the release of CoA (*Bera et al., 2009*). PqsD condenses the anthraniloyl moiety with malonyl-CoA to form 2,4-Dihydroxyquinoline (DHQ), the only terminate non-alkylated quinoline species (*Zhang et al., 2008*). Formation of 2-heptyl-4-quinolone (HHQ) requires PqsD, PqsB, and PqsC to condense longer chain fatty acids with anthraniloyl-CoA (*Bredenbruch et al., 2005*). PqsH converts HHQ to 2-heptyl-3-hydroxy-4-quinolone (PQS) in the presence of oxygen (*Schertzer, Brown & Whiteley, 2010*). Unsaturations, different alkyl chain lengths, and modification of the ring-substituted nitrogen generate the diversity of alkylquinolones (*Deziel et al., 2004*). Together, the diversity of quinolones and their effects on *P. aeruginosa* pathogenicity is still not well understood.

In addition to its functions in cell-to-cell communication, PQS exhibits other biological activities that are also important in *P. aeruginosa* virulence (*Dubern & Diggle, 2008*; *Williams & Camara, 2009*). However, in hypoxic environments such as the thick

mucus of the CF lungs, PQS production ceases due to the lack of oxygen (*Schobert & Jahn, 2010*; *Su & Hassett, 2012*). In contrast, DHQ is the most abundant quinolone in *P. aeruginosa* planktonic cultures and, unlike PQS, its synthesis does not require oxygen (*Zhang et al., 2008*). Our previous work showed that mutants only producing DHQ maintained pyocyanin production and antifungal activity of *P. aeruginosa* (*Rella et al., 2012*). However, the role of DHQ is not well understood and its determination may further help understand the regulatory activities within the Pqs system and identify new strategies for anti-*Pseudomonas* treatments. In this study, we investigated the mechanism by which DHQ affects *P. aeruginosa* virulence, regulates *pqs* operon transcription through PqsR, and interacts with PqsR. To mimic the environment colonized by *P. aeruginosa* during chronic lung infections, we also determined the effect on quinolone composition during growth with different amounts of available oxygen and nutrients. Finally, we assessed CF patient sputum samples for quorum-sensing molecules by mass spectrometry analysis. Our findings suggest that DHQ plays a role in *P. aeruginosa* pathogenicity and may help to establish *P. aeruginosa* infection under oxygen-limiting conditions.

## METHODS

### Bacterial strains, plasmids, and media

*P. aeruginosa* wild-type strain PAO1 and its derived mutants, and *E. coli* strains were grown in Luria–Bertani (LB) medium at 37 °C in a shaker incubator. Cystic fibrosis mimic media was prepared according to *Palmer et al. (2005)*. Hypoxic conditions were generated by flushing media with $N_2$ for 10 min and incubating cultures in a screw-cap vial sealed with a Tephlon© insert and nitrogen-filled head space. Antibiotic concentrations for bacterial selection were: 50 µg/ml and 200 µg/ml carbenicillin (CBC) for *E. coli* and *P. aeruginosa*, respectively, 30 µg/ml kanamycin (Kan), 34 µg/ml chloramphenicol (Cam), 30 µg/ml gentamicin (Gm). Culture density was assessed by measuring absorbance at 600 nm. DHQ (2,4-dihydroxyquinoline) and farnesol were purchased from Sigma-Aldrich and PQS (2-heptyl-3-hydroxy-4-quinolone) and HHQ (2-heptly-4-quinolone) were purchased from Qingdao Vochem Co.

### Generation of mutants

*P. aeruginosa* mutant strains Δ*pqsAB*, Δ*pqsB*, and Δ*pqsC* were generated by homologous recombination using a protocol described previously (*Choi & Schweizer, 2005*). The mutant alleles were constructed by overlapping PCR to contain a gentamicin-resistance cassette flanked by 5′ and 3′ fragments of the gene to be deleted. The mutant fragments were inserted into pEX18ApGW, a suicide vector, to produce gene knockout plasmids. Each knockout plasmid was transformed into *E. coli* strain SM10 and conjugally transferred from SM10 to PAO1. The resultant integrants were selected on PIA medium containing 30 µg/ml gentamicin (Gm30). To resolve merodiploids, Gm-resistant colonies were streaked for single colonies on LB+Gm30 plates containing 5% sucrose. The unmarked deletion mutants were generated by Flp-mediated marker excision utilizing pFLP2. Potential mutants were screened by PCR using corresponding flanking primers and were confirmed by sequencing.

## Generation of protein-expressing plasmids and protein purification

Coding sequences of PqsR and PqsR-C[87], a truncated version of PqsR containing the ligand-binding domain starting from Cys87, were amplified by PCR and cloned into a modified pET-28a vector to express recombinant PqsR and PqsR-C[87] as His-tagged SUMO fusion proteins for increased solubility. *E. coli* strain BL21(DE3) or Rosetta containing the expression plasmids was grown in LB with antibiotics (Kan for BL21(DE3), Kan and Cam for Rosetta) to $OD_{600}$ of 0.6 and heat-shocked at 45 °C for 45 min. Overexpression of recombinant protein was induced by 0.1 mM IPTG at 16 °C for 16 h. The cells were harvested by centrifugation, resuspended in 20 mM Tris–HCl (pH 8.0) with 500 mM NaCl, and lysed by sonication. His-tagged protein in the cell-free extract was purified by nickel affinity chromatography. Fractions containing pure protein were pooled, concentrated, and stored at −80 °C. Protein concentration was determined using the Bradford method with $\gamma$-globulin as the protein standard.

## *Caenorhabditis elegans* survival and imaging

Bristol N2 strain of *C. elegans* was maintained and synchronized on nematode growth media (NGM) containing *E. coli* OP50 according to the WormBook (*Girard et al., 2007*). Synchronized L4 nematodes (around 30 per plate) were transferred to 60 mm petri dishes containing a lawn of the tested *P. aeruginosa* strain. The antimetabolite 5-fluorouracil (25 µM) was added to the NGM agar to inhibit worm reproduction during the course of the experiment. The plates were incubated at 25 °C and worms were monitored daily for survival. Worms were determined dead when no movement was observed following stimulation with a platinum wire. To assess *in vivo* bacterial colonization, *P. aeruginosa* strains PAO1, ΔpqsAB, and ΔpqsB carrying the GFP-expressing plasmid pSMC2 (*Bloemberg et al., 1997*) were used for infection. At the indicated time-points, nematodes were transferred onto fresh agar to remove free-living bacteria, placed into 10 µl of water on a glass slide, and viewed using a Nikon TE2000-S Epifluorescent microscope with a CRI-Nuance imaging system. Images were color-enhanced and analyzed with the ImageJ software.

## Pyocyanin quantification

Pyocyanin concentration was measured using a modified protocol from previous work (*Das & Manefield, 2012*). Briefly, 400 µl of cell-free supernatant from an overnight culture was mixed vigorously with 240 µl chloroform. After organic-phase separation, 200 µl of the organic phase was transferred to a new tube and mixed with 60 µl 0.2 N HCl. Pyocyanin was measured spectrophotometrically at 520 nm using a NanoDrop ND-1000 spectrophotometer.

## Quantitative real-time PCR (qRT-PCR)

*P. aeruginosa* strains were grown in LB until early stationary phase. Total RNA was extracted, treated with DNaseI, and precipitated overnight at −20 °C. cDNA was generated using the Bioline Tetro cDNA synthesis kit. Real-time quantitative PCR was performed on a Bio-Rad MyiQ single-color real-time PCR detection system. The primers used for each gene were as follows: for *pqsA*, forward primer (TTCTGTTCCGCCTCGATTTC)

and reverse primer (AGTCGTTCAACGCCAGCAC); for *rpoD*, forward primer (GGGC-GAAGAAGGAAATGGTC) and reverse primer (CAGGTGGCGTAGGTGGAGAAC). PCR reactions were performed in a 96-well plate with a reaction mixture containing 10 µl SensiFAST SYBR Green mix (Bioline Reagents, London, UK), 200 ng cDNA, and 200 nM each of forward and reverse primers in a total volume of 20 µl. All reactions were performed in triplicate. The thermal cycling conditions were set at 95 °C for 3 min, followed by 40 cycles of 2-step amplification (10 s at 95 °C and 45 s at 57 °C). Data were analyzed with MyiQ software. The abundance of *pqsA* gene was normalized against the house-keeping *rpoD* gene using the $\Delta\Delta Ct$ method.

### *pqsA'*-LacZ fusion reporter assay

LacZ reporter assay was performed in both *E. coli* and in *P. aeruginosa* strains as previously described (*Cugini et al., 2007*). Briefly, overnight culture of *E. coli* containing the pEAL08-2 reporter construct was diluted to $OD_{600nm}$ of 0.05, grown in the presence of quinolones for 2 h, and harvested by centrifugation. *P. aeruginosa* strains with pEAL08-2 were grown overnight before harvest. Cells were lysed and the LacZ activity was measured using a $\beta$-galactosidase enzyme assay kit (Promega, Madison, WI, USA). The LacZ product was measured at 420 nm on a plate reader (BioTek Synergy HT; BioTek, Winooski, VT, USA).

### Electrophoretic mobility shift assay (EMSA)

EMSA assays were performed using cell lysates from PAO1, *pqs* mutants, and Rosetta strains induced for His-SUMO or His-SUMO-PqsR production as described (*Cugini et al., 2007*). For each sample, 10 µg of cell-lysate was incubated with 0.15 fmol of biotinylated 248-bp *pqsA* promoter DNA (*pqsA'*), which was generated by PCR using biotinylated forward primer (TTCTTGCTTGGTTGCCG) and reverse primer (GACAGAACGTTCCCTCTT). Unlabeled *pqsA'* probe was generated using the same primers as the biotinylated-*pqsA'* without the biotin modification. The reaction mixture contained 10 mM Tris–HCl (pH 8.0), 1 mM EDTA, 50 mM NaCl, 1 mM DTT, 1 µg/µl Poly(dI-dC) and was incubated at room temperature (24 °C) for 20 min. Samples were separated on a 5% polyacrylamide gel in 0.5×Tris Borate EDTA (TBE) run at 100 V for 50 min and transferred onto a positively-charged nylon membrane at 40 V for 1 h. Biotinylated DNA on the nylon membrane was probed by streptavidin-HRP conjugate, detected by a chemiluminescent substrate (Pierce LightShift Chemiluminescent EMSA kit; Thermo Fisher Scientific, Waltham, MA, USA), and visualized by exposing to either an X-ray film or GE ImageQuant RT-ECL.

### Saturation transfer difference (STD) NMR

STD NMR experiments were prepared with 1 µM SUMO-PqsR-C[87] with increasing concentrations of DHQ (10 nm to 100 µM). Data were collected at 298 K on a Bruker Avance III 600 MHz NMR spectrometer equipped with a 5 mm cryogenically-cooled QC-Inverse probe and using a standard STD pulse sequence with 30 ms 8.4 kHz spin lock to minimize background protein resonances (*Kemper et al., 2010*). Solvent suppression was achieved using the excitation sculpting scheme. Saturation of the protein signals was

performed using a train of 10, 20, or 59 selective 56 dB Gaussian pulses of 50 ms duration with total saturation times of 0.25 s to 5.0 s. The on-resonance frequency was set up at −0.5 ppm. STD spectra were acquired from 64 scans, 2,050 receiver gain, and 14 ppm sweep width.

## Quantification of quorum-sensing molecules

Supernatants of overnight cultures of PAO1 grown aerobically or anaerobically were collected and acidified with 0.1% formic acid. RP-HPLC coupled with a mass detector was used to analyze and quantitate QS molecules in culture supernatants. The HPLC-MS system used was a Waters HPLC system equipped with a Waters ZQ Single Quadrapole Mass Detector outfitted with MASS LYNX software. The HPLC/MS method was water/acetonitrile (ACN) gradient with 0.1% formic acid in both solvents. Samples (10 µl) were loaded onto an Ascentis Express C18 column (Sulpeco Analytical, 5 µm particle size, 150 mm × 2.1 mm) and eluted using an acetonitrile (ACN) gradient at a flow rate of 0.4 ml/min: 30% ACN for 2 min, a linear gradient from 30 to 100% ACN over 30 min, 100% ACN for 5 min. Column temperature was held at 30 °C. The concentrations of QS molecules were determined using calibration curves generated with commercial chemical standards at concentrations ranging from 0.1 µM to 10 mM.

## Patient recruitment

This study protocol was approved by the Institutional Review Board of the Medical University of South Carolina, and the study was carried out according to the approved guidelines (IRB number: Pro00012798). Written informed consent was obtained from each patient at the time of recruitment. Patients were selected using the following criteria: over 18 years old, had a positive culture history of *P. aeruginosa* but negative for *Burkholderia* species, non-smoker, readily produced sputum, and received routine treatment at the Adult CF Center of the Medical University of South Carolina. Sputum samples were collected during patients' routine visits at the CF Center clinic and stored at −80 °C until the samples were analyzed.

## Quantification of QS molecules in sputum samples

Sputum samples were thawed and digested with 10% sputolysin (Calbiochem, San Diego, CA, USA) in PBS at 37 °C for 15 min. Digested samples were extracted with 3 ml acidified ethyl acetate (0.1% formic acid) twice. The organic layers were pooled and dried under nitrogen. Samples were dissolved in 500 µl methanol and 10 µl sample was analyzed by the HPLC-MS as described above to determine the concentrations of QS molecules in the sputum.

## Statistical analysis

The Log-Rank (Mantel-Cox Test) test was used to analyze the *C. elegans* survival data. A $p$-value of less than or equal to 0.05 was considered as a significant difference. Other data was analyzed using the Student's $t$-test (DF = 1), significant differences were identified with a $p$ value of less than or equal to 0.05.

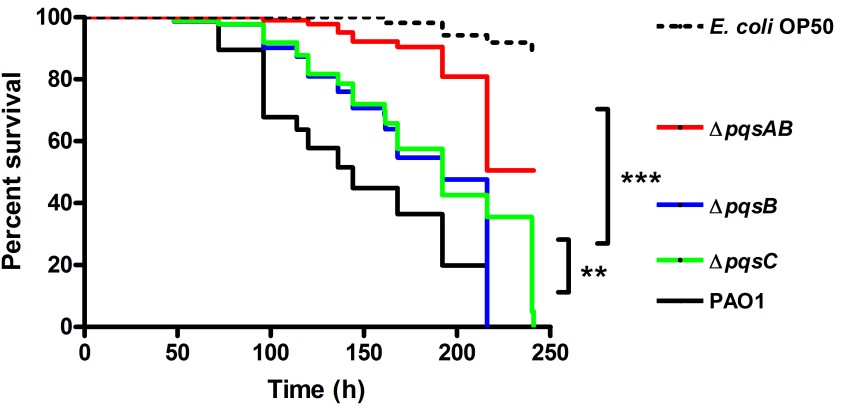

**Figure 1** **Survival of *C. elegans* infected with PAO1 and *pqs* mutants.** *P. aeruginosa* strains were incubated with *C. elegans* and worm survival was monitored daily for over 10 days. Each assay contained 20–30 nematodes per strain and was performed in triplicate. Survival data of three independent experiments were combined and plotted in a Kaplan–Meier survival curve. *E. coli* strain OP50 was used as a non-pathogenic control. Number of worms used for each strains: PAO1 $n = 77$, *pqsAB* mutant $n = 113$, *pqsB* mutant $n = 89$, *pqsC* mutant $n = 93$, *E. coli* OP50 $n = 115$. The survival data were analyzed using the Log-Rank test. Significant *p*-vulues were indicated by asterisks (**$p < 0.001$, ***$p < 0.0001$).

## RESULTS

### DHQ maintained *P. aeruginosa* virulence in *C. elegans*

*C. elegans* is an established infection model to study bacterial pathogenicity in research labs (*Korgaonkar et al., 2013*; *Kwak, Jacoby & Hooper, 2013*). *P. aeruginosa* infections are lethal to the worms due to toxic virulence factors such as pyocyanin and hydrogen cyanide, chelation of iron by PQS, and colonization of the intestinal lumen. To determine the effect of DHQ on *P. aeruginosa* pathogenicity, we monitored the survival of *C. elegans* infected with wild-type *P. aeruginosa* strain PAO1 and *pqs* mutants (Fig. 1). *E. coli* OP50 is a food source to maintain the nematodes and served as a non-pathogenic control in our study. PAO1 requires both virulence factor production and colonization of the nematodes to kill *C. elegans* following several days of incubation ($>200$ h) (*Mahajan-Miklos et al., 1999*; *Tan, Mahajan-Miklos & Ausubel, 1999*; *Tan et al., 1999*). The Pqs system regulates virulence factor production and plays a role in biofilm formation, both of which play a role in *C. elegans* killing. Using the DHQ-only producing mutant, Δ*pqsB* and Δ*pqsC* strain (*Zhang et al., 2008*), we were able to examine the effect of DHQ production on virulence compared to a quinolone-null mutant, Δ*pqsAB*.

None of the nematodes survived infections from PAO1, Δ*pqsB* or Δ*pqsC*, which only produces DHQ, after 220 h of incubation, while 50% of the nematodes infected with Δ*pqsAB* remained viable (Fig. 1). The quinolone-null mutant Δ*pqsAB* used in this study was also used in our previous work (*Rella et al., 2012*), which lacks an internal promoter of the *pqs* operon. The reduction in virulence of the Δ*pqsAB* mutant was attributed to the complete loss of quinolone production, while production of DHQ in the Δ*pqsB* and Δ*pqsC* mutant maintained pathogenicity towards the nematodes. PqsB and PqsC are both essential for alkylquinolone synthesis, but are not required to produce DHQ (*Zhang et al., 2008*). *C. elegans* killing by the Δ*pqsB* and Δ*pqsC* mutants showed that DHQ production

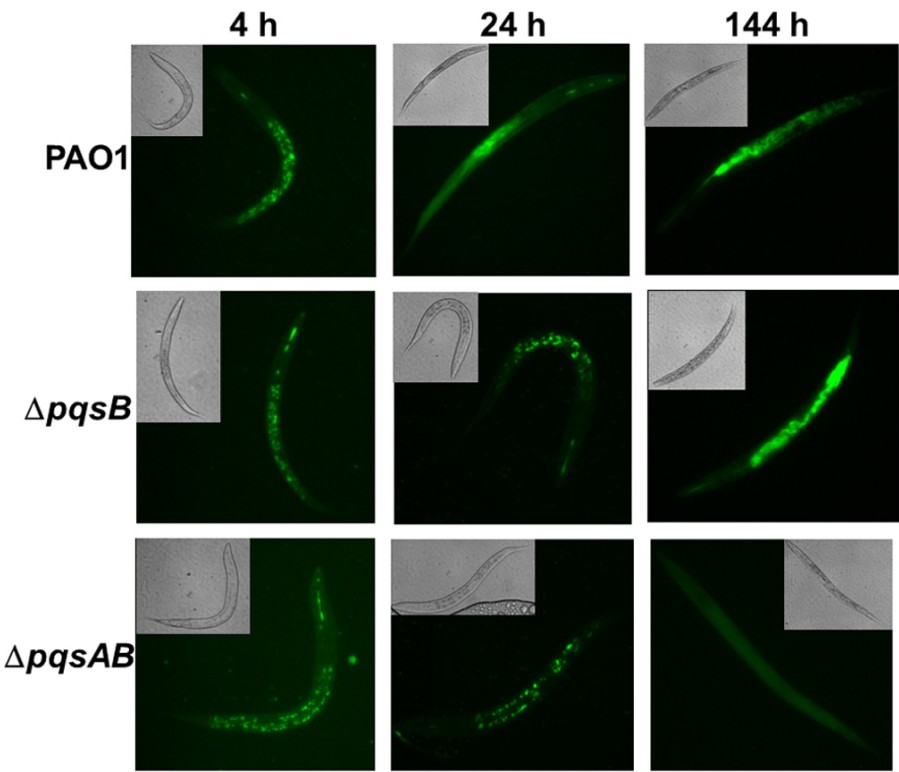

**Figure 2 Colonization of *C. elegans* by *P. aeruginosa* strains expressing GFP.** Live nematodes were selected to monitor fluorescence from the bacteria. Excitation at 435 nm for 5 s was used for samples from 4 and 24 h, while excitation for 1 s was used at 144 h. Emission was set at 485 nm. 4–6 nematodes per strain were selected from each time point and images from a representative experiment were shown. The GFP-fluorescent signal was quantitated with software ImageJ and the quantified data were shown in Fig. S1.

maintained the same level of virulence in both mutants, which were more virulent against the worms than the quinolone-null mutant Δ*pqsAB*. Thus, the production of DHQ increased pathogenicity of *P. aeruginosa* compared to a quinolone-null strain.

Fluorescence-producing bacteria have been used to infect *C. elegans* to study the internalization of bacteria and their progress through the intestines (*Sifri et al., 2003*). To visualize colonization of *C. elegans* by different *P. aeruginosa* strains, we monitored nematodes infected with PAO1 and the *pqs* mutants carrying a plasmid that stably expressed GFP (*Bloemberg et al., 1997*). After 4 h of co-incubation, all nematodes examined contained GFP-expressing bacteria (Fig. 2). After 24 h, GFP-expressing bacteria were distributed throughout the intestinal tract of the nematodes. At 144 h, *C. elegans* infected with PAO1 and Δ*pqsB* showed higher GFP fluorescence than those infected with the Δ*pqsAB* mutant (Fig. 2). We determined the difference in bacterial colonization by comparing fluorescence of the three strains following 1 s of excitation versus 5 s of excitation. A 5 s excitation led to oversaturated GFP signal in nematodes infected with PAO1 and the Δ*pqsB* mutant, while minimal fluorescence was detected within the Δ*pqsAB* mutant infected worms. A 1 s excitation resolved the oversaturation in PAO1 and Δ*pqsB*- infected nematodes and resulted in only background fluorescence from nematodes infected with

the $\Delta pqsAB$ mutant (Fig. 2). The 144 h was an important time point as it coincided with a sharp decrease in *C. elegans* survival among the nematodes infected PAO1 and $\Delta pqsB$ (Fig. 1). The GFP-fluorescent signals from the infected worms were quantitated using the software ImageJ (Fig. S1). Consistent with the observations on the fluorescent images, no significant difference was detected at 4 h and 24 h post infection. At 144 h time point, worms infected with the *pqsAB* mutant exhibited significant lower GFP signal than PAO1-infected ($p < 0.05$) and *pqsB*-infected worms ($p < 0.05$). Altogether, our results indicated that production of DHQ increased the ability of *P. aeruginosa* to colonize and infect *C. elegans*, which may point to a role in causing infection of a human host.

## DHQ supported pyocyanin production *in vitro*

Virulence factor production is also important for PAO1 infection of *C. elegans* (*Eriksson et al., 2009*). Pyocyanin, a blue–green redox-active pigment, is a potent virulence factor that is toxic towards other microorganisms and eukaryotic cells (*Lau et al., 2004*; *Papaioannou, Utari & Quax, 2013*; *Rella et al., 2012*). Both PqsR and PqsE are important for Pyocyanin production in *P. aeruginosa*: PqsR activates transcription of the *pqs* operon (*Williams & Camara, 2009*), PqsE function remains enigmatic until recently being shown to have pathway-specific thioesterase activity and promote alkylquinolone signaling (*Drees & Fetzner, 2015*).

In order to demonstrate that changes in virulence of the *pqs* mutants were not due to defects in growth, the growth of the *pqs* mutants were monitored in LB and synthetic cystic fibrosis medium (SCFM) that was developed based on the nutrient composition of the CF sputum (*Palmer, Aye & Whiteley, 2007*). The growth curves of the *pqs* mutants were similar to that of the wild-type PAO1 in both media, demonstrating that these mutants were not defective in replication (Figs. 3A and 3C). Next, we quantified pyocyanin from overnight cultures of PAO1 and *pqs* mutants grown in LB media. PAO1 produced the highest amount of pyocyanin (about 2.5 μg/ml) followed by the $\Delta pqsB$ mutant, which produced 50% of the wild-type level (Fig. 3B). The $\Delta pqsAB$ mutants produced the least amount of pyocyanin (Fig. 3B). Nutrient conditions can alter bacterial phenotypes. The SCFM mimics the natural environments of *P. aeruginosa* and provides less nutrients than LB. Although reduced, a similar trend in pyocyanin production by the strains was observed compared to LB-grown cultures (Fig. 3D). These results support that DHQ production partially maintained *pqs* signaling and pyocyanin production, whilst alkylquinolones were required for full virulence production.

## DHQ activated PqsR for transcription of the *pqs* operon

PQS and HHQ activate PqsR for transcription of the *pqs* operon (*Diggle et al., 2007*; *Dubern & Diggle, 2008*). To determine if DHQ affected *pqs* operon transcription, we used qRT-PCR to monitor the expression of *pqsA* from PAO1 and *pqs* mutants using RNA polymerase sigma factor *rpoD* as the house-keeping gene (Fig. 4). $\Delta pqsB$ mutant expressed *pqsA* at 47% of the wild-type level, while $\Delta pqsAB$ mutant showed no detectable *pqsA* transcript (Fig. 4). Deletion of *pqsR* also reduced *pqsA* transcription to basal level, which was consistent with PqsR being an activator of the *pqs* operon. Deletion of both

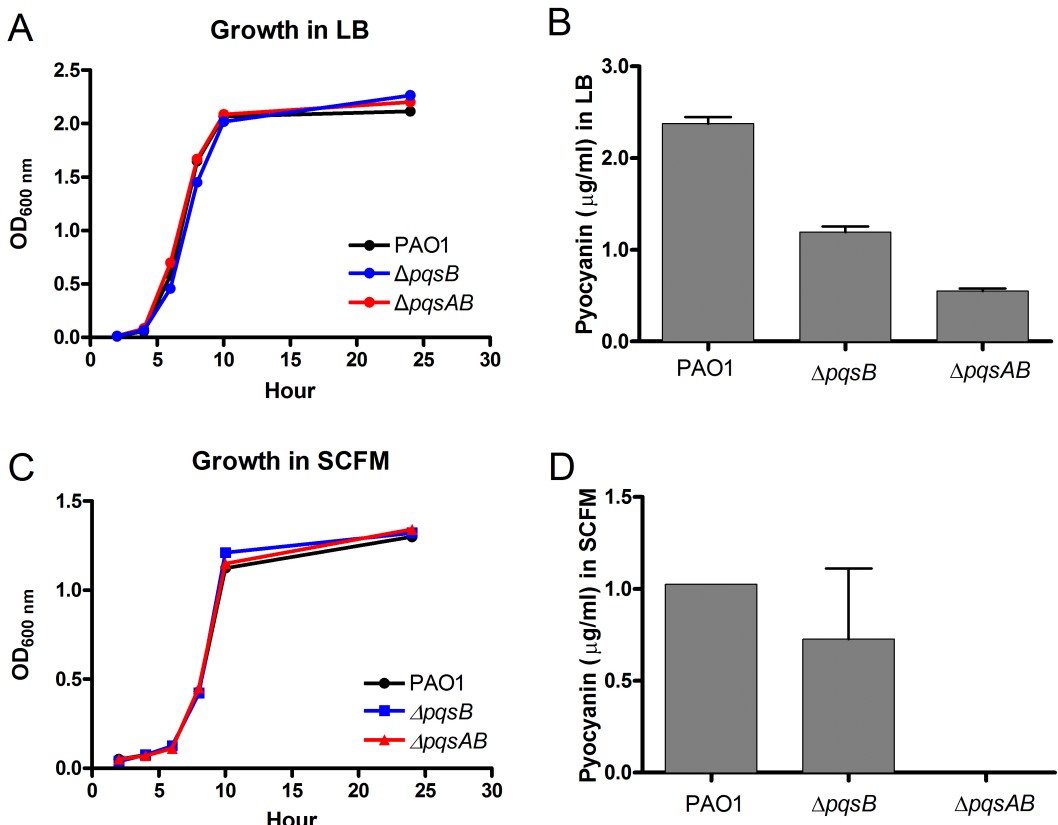

**Figure 3 Growth and pyocyanin production by PAO1 and *pqs* mutants.** (A) Growth curves of *P. aeruginosa* strains in LB over 24 h. (B) Pyocyanin was extracted from 18-h planktonic cultures grown in LB, quantified spectrophotometrically at 520 nm, presented as mean ±SE from at least three independent measurements. Student's *t*-test of *pqsB* mutant produced significantly higher level of pyocyanin than *pqsAB* mutant (*p*-value = 0.0003). (C) Growth curves of *P. aeruginosa* strains in SCFM over 24 h. (D) Pyocyanin was extracted and quantified from 24 h planktonic cultures grown in SCFM. The pyocyanin from *pqsAB* mutant in SCFM medium was below the detectable level. Experiments were performed in duplicate and data were presented as mean ± SE.

*pqsR* and *pqsB* also resulted in basal level of *pqsA* transcript (Fig. 4), suggesting that the activation of *pqs* operon transcription in Δ*pqsB* was dependent on PqsR.

To investigate the effect of exogenous and endogenous DHQ on PqsR activity, we performed promoter-fusion assays in *E. coli* and *P. aeruginosa* strains (Fig. 5). The promoter-fusion construct contained the upstream regulatory element (−500 bp) of *pqsA* fused with the coding sequence of LacZ (Cugini et al., 2007). The reporter construct also contained the *pqsR* gene, which was controlled by a *tac* promoter. DHQ supplemented to *E. coli* containing the reporter construct displayed a concentration-dependent increase in LacZ activity (Fig. 5A). 100 µM DHQ increased the *pqsA* promoter activity by 60 % over the DMSO-treated control, compared with 110% activation by 1 µM PQS (Fig. 5A), suggesting that PQS was more potent in activating PqsR. The effect of endogenous quinolones on PqsR activity was tested in PAO1 and *pqs* mutants carrying the reporter construct. The *pqsA* promoter activity from the DHQ-only Δ*pqsB* mutant

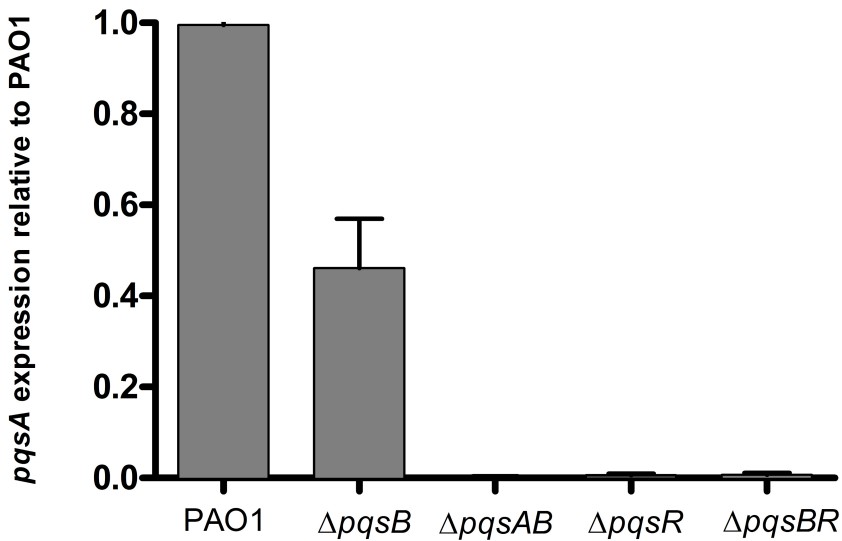

**Figure 4 Expression of *pqsA* in PAO1 and *pqs* mutants.** Strains grown in LB until cell density reached $OD_{600nm}$ of 1.8. *rpoD* was used as the house-keeping gene for normalization. Relative abundance of *pqsA* transcript from *pqs* mutants compared with the wild-type PAO1 level, which was set as 1. Experiment was performed in triplicate with data shown as mean ± SE. The Student's *t*-test analyses showed that *pqsA* expression level in the *pqsB* mutant, although lower than the wild-type PAO1 ($p$-value = 0.008), was statistically significantly higher than *pqsAB* ($p$-value < 0.0001), *pqsR* ($p$-value < 0.0001), and *pqsBR* mutants ($p$-value < 0.0001).

was similar to that of PAO1 while the Δ*pqsAB* strain displayed basal activity (Fig. 5B), indicating endogenous produced DHQ can activate PqsR binding to *pqsA* promoter.

To test if DHQ can compete with alkylquinolones for PqsR, DHQ was supplemented in the presence of HHQ or PQS in the *E. coli* reporter system. DHQ supplemented with either HHQ or PQS did not significantly alter the *pqsA* promoter activity compared to HHQ and PQS alone (Fig. 5C). Various fungal species produce farnesol as a quorum-sensing molecule, which is also an antagonist of PqsR (*Cugini et al., 2007*). In the *E. coli* reporter system, farnesol reduced LacZ activity below the background level (Fig. 5D), consistent with farnesol as an inhibitor of PqsR. Interestingly, co-treatment of farnesol and DHQ alleviated the inhibition of PqsR by farnesol, albeit at a lower level compared with PQS (Fig. 5D). Taken together, endogenous transcription of *pqsA* from PAO1 and *pqs* mutants, as well as activity from exogenous DHQ supplemented to the reporter system, show that DHQ can function as a ligand of PqsR to activate the transcription of the *pqs* operon.

### *In vitro* studies on the interaction of DHQ with PqsR

The interactions between alkylquinolones with PqsR are demonstrated by a recently solved crystal structure of PqsR complexed with its ligand, which shows how ligand binding affects PqsR affinity for its DNA-binding site (*Ilangovan et al., 2013*). To test how DHQ binding affects PqsR interaction to its DNA binding site, we performed

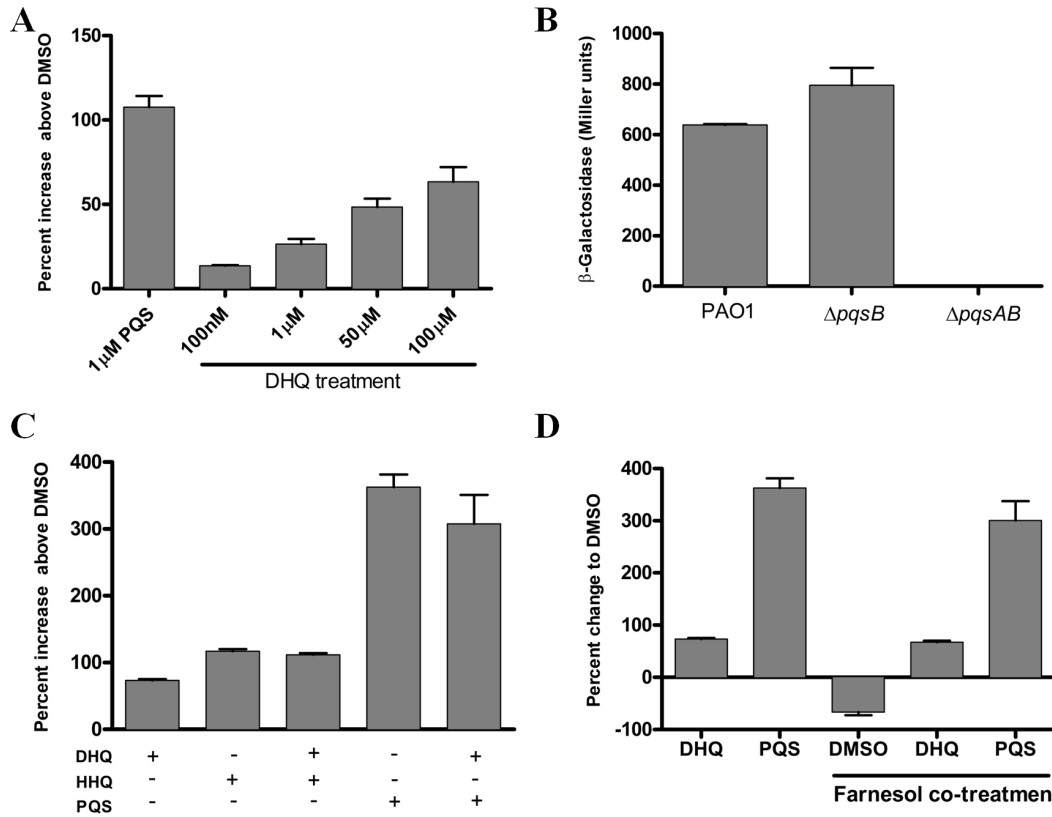

**Figure 5** **Reporter assay of PqsR activity in *E. coli* and *P. aeruginosa*.** (A) DHQ supplemented to *E. coli* carrying the reporter construct pEAL08-2. LacZ activity in cells treated with DMSO was set as the baseline. DHQ increased PqsR activity in a dose-dependent manner whilst PQS was more potent in activating PqsR ($p$-value $= 0.013$ from comparing 1 $\mu$M PQS and 100 $\mu$M DHQ by the Student's $t$-test). (B) Overnight cultures of PAO1 and *pqs* mutants containing the reporter plasmid were used to determine the effect of endogenously generated DHQ on PqsR activation of *pqsA* promoter. PAO1 and *pqsB* mutant exhibited similar PqsR activity whereas no detectable activity was observed in the *pqsAB* mutant, suggesting that endogenously produced DHQ from the *pqsB* mutant was sufficient to maintain PqsR activity in the absence of alkylquinolones. (C) Co-supplementation of 100 $\mu$M DHQ with 30 $\mu$M HHQ or 30 $\mu$M PQS in *E. coli* reporter. No statistically significant difference was observed between alkylquinolone (HHQ or PQS) only group and alkylquinolone plus DHQ (HHQ + DHQ or PQS + DHQ) group. (D) Competition of 100 $\mu$M DHQ or 30 $\mu$M PQS with 250 $\mu$M farnesol in the *E. coli* reporter. DHQ significantly abolished the inhibition of PqsR activity by farnesol ($p$-value $= 0.0023$ between DHQ + farnesol treatment and DMSO + farnesol treatment by the Student's $t$-test). Data from independent experiments were presented as mean $\pm$ SE and analyzed by the nonparametric Student's $t$-test.

electrophoretic mobility shift assays (EMSA) with biotinylated-*pqsA* promoter (Bio-*pqsA*) and cell lysates of PAO1 and the *pqs* mutants (Fig. 6). Only PAO1 and $\Delta pqsB$ cell lysates shifted Bio-*pqsA* migration, while the $\Delta pqsAB$ and $\Delta pqsR$ mutants did not show any effect (Fig. 6A). Addition of unlabeled *pqsA* probe to PAO1 and $\Delta pqsB$ cell lysates decreased the amount of shifted Bio-*pqsA* probe, demonstrating the interaction was specific for *pqsA* promoter sequence (Fig. 6B). Recombinant PqsR as a fusion protein with N-terminal His-tagged SUMO also showed increased shifted *pqsA* probe in the presence of PQS or DHQ (Fig. 6C). No *pqsA* probe was shifted in the control lysate containing His-SUMO tag, demonstrating the shifted *pqsA* probe was due to PqsR activity specifically

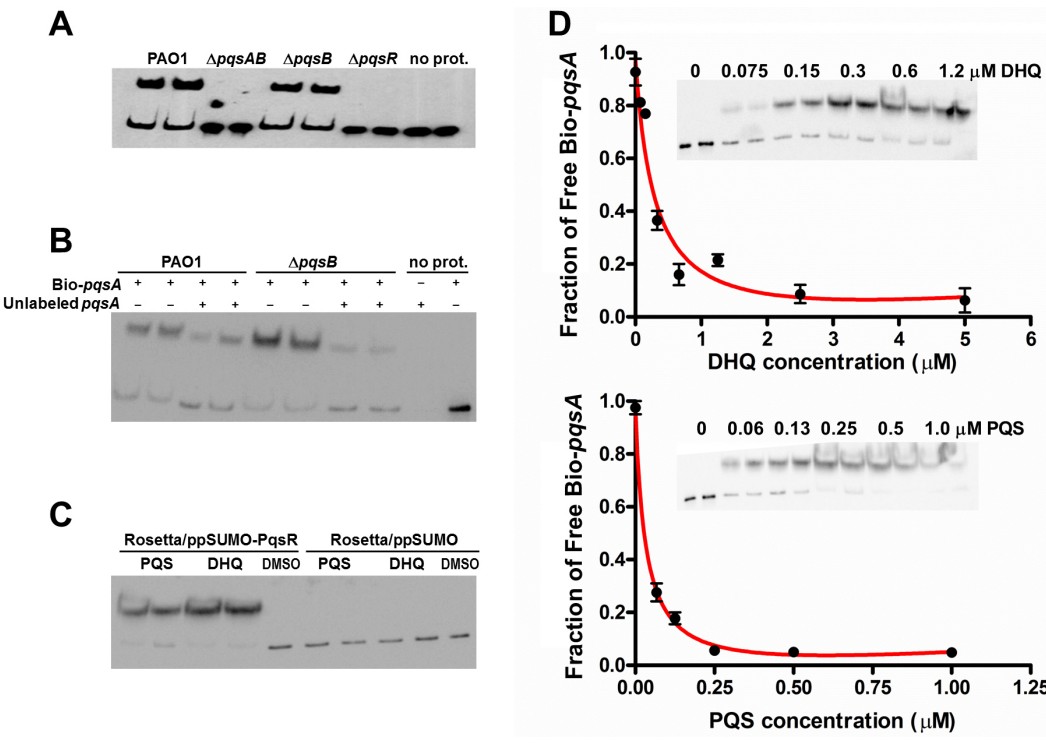

**Figure 6** **DHQ promoted PqsR binding to *pqsA* promoter.** (A) EMSA assays with Biotinylated-*pqsA* (Bio-*pqsA*) probe incubated with cell-lysates of PAO1 and *pqs* mutants. (B) The interaction was specific for *pqsA* promoter as unlabeled-*pqsA* competed with Bio-*pqsA* probe. (C) Both DHQ (200 μM) and PQS (30 μM) promoted the interaction between recombinant PqsR with Bio-*pqsA*. (D) DHQ (0–5 μM) and PQS (0–1 μM) promoted recombinant PqsR binding to Bio-*pqsA* probe in a concentration dependent manner. Quantification of free Bio-*pqsA* was performed using ImageJ software and non-linear curve fitting was done using GraphPad Prism software to determine the apparent Kd of PqsR for DHQ and PQS.

(Fig. 6C). The affinity of PqsR for DHQ and PQS was determined semi-quantitatively by EMSA by measuring free Bio-*pqsA* in response to quinolone supplemented at different concentrations. The calculated Kd of PqsR for DHQ was 150 nM and 33 nM for PQS (Fig. 6D).

Saturation transfer difference NMR (STD-NMR) detects transient binding of ligand to a protein in solution and was used to study the interaction between DHQ and PqsR. The ligand binding domain of PqsR starting from residue Cys87 (PqsR-C87) was purified as a His-tagged SUMO fusion protein and used in the STD-NMR experiments (Fig. 7). Novel peaks on the PqsR-C87 spectra were quantified for intensity and modeled for single-site binding kinetics (Fig. 7A). The Kd was estimated to be 450 nM (Fig. 7B). Interestingly, the saturation time showed a strong interaction after 3 s, while exhibiting fast-exchange interactions from 0.5 to 2.5 s. The fast-exchange may be explained by hydrophobic interactions within the hydrophobic pocket of PqsR until DHQ was coordinated through hydrogen binding. Using the D-COSY assignments, we mapped the 1-D hydrogen spectra to show that the meta-hydrogen of DHQ participated in hydrogen-binding with PqsR (Fig. 7C). The ortho- and para-hydroxyl groups were not resolved from the water peak.

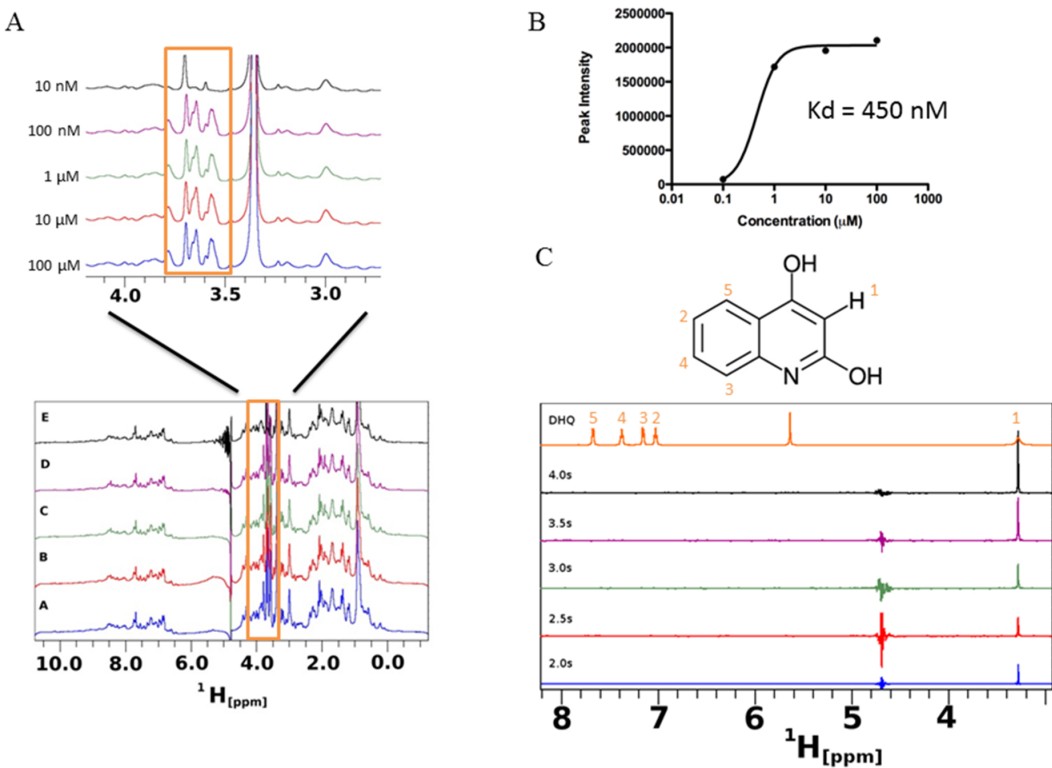

**Figure 7  Saturation transfer difference (STD) NMR of SUMO-PqsR-C[87] interaction with DHQ.** (A) STD-NMR spectra of 1 µM SUMO-PqsR-C87 titrated with 10 nm to 100 µM DHQ. A close-up view of the changing spectra was used to highlight the differences in peak intensities following increased DHQ concentrations. (B) Peak intensity of the most prominent peak (close-up view) versus concentration of DHQ was used to determine Kd of binding. (C) STD-NMR of 100 µM DHQ added to 1 µM SUMO-PqsR-C87 and measured for saturation times in 0.5 s intervals. Intervals selected to show saturation from initial incubation to 5 s. D-COSY assignments of DHQ used to map interacting hydrogens (numbered in orange).

However, we predict those groups were also participating in hydrogen binding based on preliminary docking analyses.

## DHQ production under anaerobic conditions

Our results suggested DHQ, similar to PQS and HHQ, promotes *P. aeruginosa* pathogenesis by activating PqsR; however, conditions found in CF lungs may limit PQS production due to the lack of oxygen (*Schobert & Jahn, 2010*; *Worlitzsch et al., 2002*). Further complicating alkylquinolone syntheses is the reduced metabolism in *P. aeruginosa* biofilms limiting the availability of fatty acid substrates from $\beta$-oxidation. As DHQ formation requires smaller, less energy intensive precursors and does not rely on oxygen in the environment, DHQ may play a role apart from the alkylquinolones during chronic colonization of the CF lung.

Strict anaerobic cultures do not produce PQS and express lower *pqs* operon activity compared to aerobic cultures (*Lee et al., 2011*). To determine how DHQ concentration is affected during anaerobic growth, we quantified QS molecules from PAO1 cultures grown

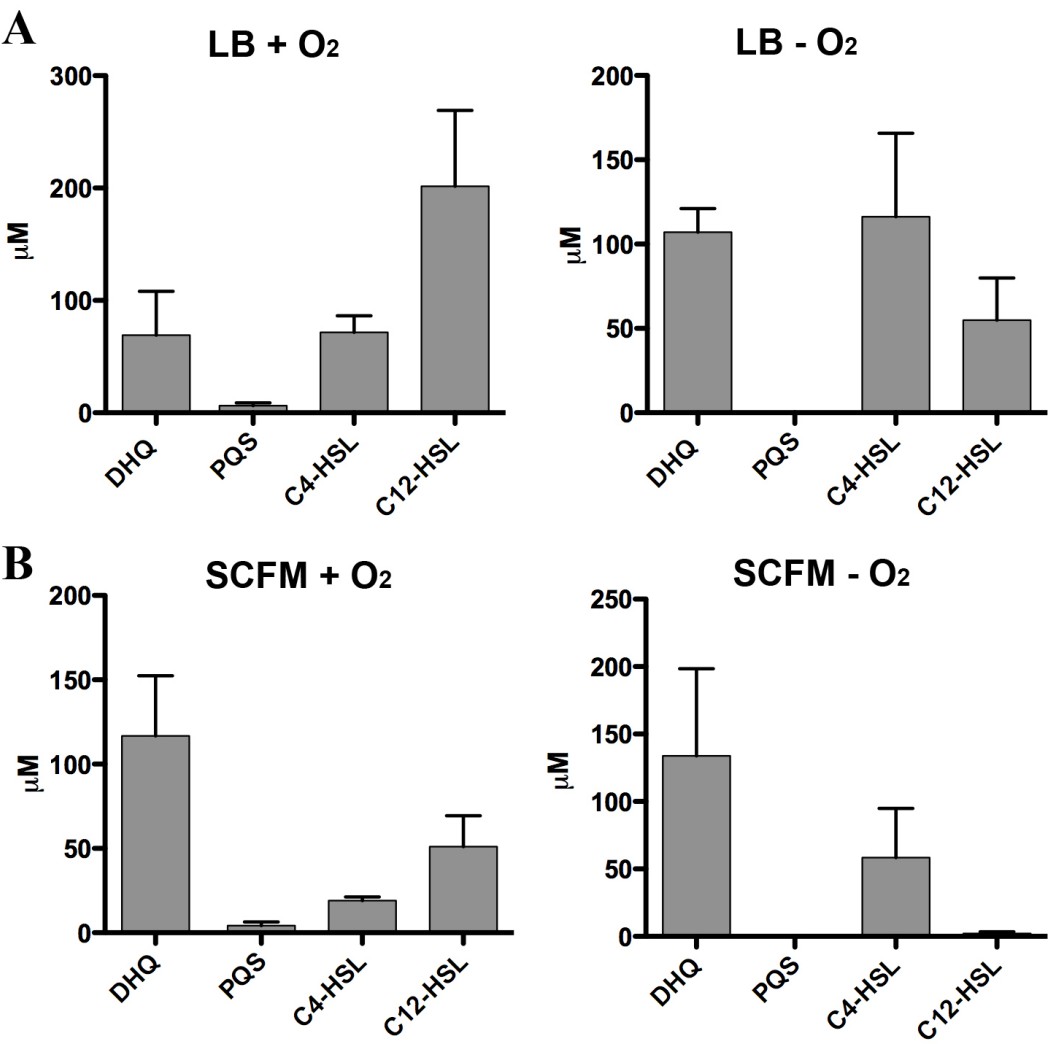

**Figure 8** **Quantification of extracellular levels of DHQ from aerobic and anaerobic PAO1 cultures.** (A) PAO1 cultures were grown aerobically in LB and anaerobically in LB+400 $\mu$M sodium nitrate until the culture reached $OD_{600}$ of 2.0. Culture supernatants were acidified with formic acid and QS molecules were analyzed using HPLC-MS. The concentrations were determined using calibration curves of QS standards. (B) PAO1 cultures in SCFM were treated in the same manner and grown aerobically and anaerobically to determine the concentrations of QS molecules. Data represented as mean $\pm$ SE from at least two independent experiments.

aerobically and anaerobically (Fig. 8). Concentration of $C_4$-HSL was higher than 3-oxo-$C_{12}$-HSL in the anaerobic samples, consistent with the relative abundance of the two HSL signals under anaerobic conditions as found in the CF lungs (*Hassett et al., 2009*). High levels of extracellular DHQ were detected from cultures grown in LB both in the presence and absence of oxygen (Fig. 8A). Interestingly, anaerobic culture supernatants contained higher concentrations of DHQ compared to aerobic cultures in LB. As expected, PQS was not detected under anaerobic conditions. Similar trends of QS molecules were observed from PAO1 cultured in SCFM aerobically and anaerobically (Fig. 8B). Comparison of the DHQ concentrations between LB and SCFM showed that DHQ level was higher in the

SCFM, but the difference was not statistically significant (Fig. 8). Together, these results showed DHQ was the predominant quinolone produced in both rich and SCFM medium regardless of the oxygen level. The production of DHQ in anaerobic condition suggest a specific function of DHQ under low oxygen environments where PQS production is limited; however, further research is needed to understand *P. aeruginosa* transcriptional regulation in hypoxic conditions and the dynamics of QS molecules within the anaerobic mucous.

### Detection of DHQ in CF patient sputa

Alkylquinolones have been detected in the CF patient samples (*Collier et al., 2002*). We analyzed the sputum samples from stable CF patients to determine whether DHQ was detectable in the CF lungs. Sputum samples were collected during CF patient routine clinic visits and were analyzed by HPLC-MS for DHQ concentrations. More than 80 sputum samples from 45 patients were collected and analyzed. DHQ was readily detected in 34 sputum samples, and the concentrations covered a wide range. The median of DHQ concentration in stable CF sputa was 123 $\mu$M with an interquartile range between 25 and 464 $\mu$M. The wide range of DHQ concentrations in the patient samples reflected the non-uniform nature of the sputum samples. Nonetheless, the presence of DHQ in the patient samples demonstrated that DHQ was produced during an infection in the CF lungs, and the effects of long-term exposure to high concentration of DHQ on host cells remains to be determined.

## DISCUSSION

Alkylquinolones are well characterized for their role in activating the Pqs signaling system as a ligand of the transcription factor PqsR. Our work provides the first evidence that DHQ also binds to PqsR to activate the transcription of the *pqs* operon. Transcriptional studies and EMSA experiments demonstrated that DHQ, though not as potent as PQS, activated PqsR for increased transcription of the *pqs* operon. STD-NMR determined the interaction between DHQ and PqsR as well as the binding affinity *in vitro*. HPLC-MS analyses of *P. aeruginosa* culture supernatants showed that DHQ was present in high extracellular concentrations in both rich medium and SCFM regardless of oxygen availability. Within chronically infected patient samples, DHQ was detected in the CF patient sputum, suggesting DHQ as an extracellular effector to aid *P. aeruginosa* in the disease environment. Finally, we have established that production of DHQ maintained pathogenicity using the *C. elegans* infection model. Specifically, production of DHQ resulted in increased colonization and pyocyanin production compared with the quinolone-null mutant.

Among the three well-known QS systems in *P. aeruginosa*, the Pqs system was most recently identified (*Pesci et al., 1999*). Some aspects of the Pqs system remain to be elucidated, including the activities of specific enzymes in the biosynthesis of alkylquinolones (*Drees & Fetzner, 2015*; *Dulcey et al., 2013*) and the transcriptional regulation of the *pqs* operon (*Brouwer et al., 2014*; *Dotsch et al., 2012*; *Knoten et al., 2014*). Four transcriptional start sites (TSSs) have been identified in the *pqs* operon, two in the promoter region of

*pqsA* (*pqsA*$_{-339}$ and *pqsA*$_{-71}$), the third upstream of *pqsB* (*pqsB*$_{-31}$) (*Dotsch et al., 2012*), and the fourth just upstream of the *pqsC* stop codon (*Knoten et al., 2014*). The Bio-*pqsA* probe used in this study corresponded to the upstream segment of *pqsA*$_{-71}$ TSS. Both DHQ and PQS can bind to PqsR to activate the *pqs* operon transcription from the *pqsA*$_{-71}$ TSS. RhlR was recently shown to regulate the longer transcript of the *pqs* operon from the *pqsA*$_{-339}$ TSS by forming a hairpin structure in the 5′-untranslated sequence (*Brouwer et al., 2014*). Nothing is known about the regulation of the two internal TSSs. Future work is needed to identify transcriptional regulators of the alternate transcripts of the *pqs* operon. A putative regulatory site was identified in the second internal transcript (*Knoten et al., 2014*) and it will be interesting to determine if DHQ or alkylquinolones function as ligands for such regulators.

Apart from its role in transcriptional regulation, the importance of DHQ under anaerobic conditions should garner interest because low-oxygen condition is encountered by colonizing *P. aeruginosa* during a chronic infection. Our results showed that DHQ was secreted in high concentration regardless of oxygen availability. For anaerobic growth conditions, nitrate acts as an alternate electron acceptor in *P. aeruginosa* (*Hassett, 1996*); however, PQS inhibits nitrate-respiratory chain activity through iron chelation (*Toyofuku et al., 2008*). PQS is secreted via membrane vesicles, which are formed under both aerobic and denitrifying condition (*Toyofuku et al., 2013*). HHQ is secreted by the RND-type efflux pump MexEF-OprN, which is up-regulated under anaerobic conditions (*Fetar et al., 2011*; *Lamarche & Deziel, 2011*). Our preliminary work has also indicated that DHQ was secreted from the same efflux pump. This mechanism of secretion may therefore explain why DHQ was present in high levels in the oxygen-limiting CF sputum. As CF patients are chronically infected by *P. aeruginosa*, more work is required to determine the effects of long-term exposure of lung cells to DHQ.

Taken together, our findings support that DHQ plays similar multifactorial roles compared to the alkylquinolones in *P. aeruginosa* pathogenicity. DHQ is present at high concentrations in both aerobic and anaerobic conditions, which is significant given the nature of low-oxygen in the lungs of CF patients. However, we have only begun to understand how the quinolones function individually. Future efforts should focus on how DHQ interacts with other quinolones and extracellular molecules and how these interactions affect *P. aeruginosa* pathogenicity. It will also be interesting to determine how DHQ affects the fitness of *P. aeruginosa* in its natural environments such as soil and water and how DHQ affects other microorganisms cohabitating with *P. aeruginosa*.

## ACKNOWLEDGEMENTS

We would like to thank Dr. Roberto Kolter (Harvard University) for the plasmid pSMC2 and Dr. Deborah Hogan (Dartmouth University) for the plasmid pEAL08-2.

### Funding

This work was supported in part by the Cystic Fibrosis Foundation (ZHANG12I0), by the COBRE in Lipidomics and Pathobiology at the Medical University of South Carolina (NIH P20 RR017677), by the DOD/DMRDP (DM090161), and by the South Carolina Clinical & Translational Research (SCTR) Institute, with an academic home at the Medical University of South Carolina (NIH/NCATS UL1 TR000062). JDG was supported by the South Carolina Clinical & Translational Research (SCTR) Institute, with an academic home at the Medical University of South Carolina, via the SCTR Predoctoral Clinical & Translational Research Training Program (NIH/NCATS TL1 TR000061). The funders had no role in study design, data collection and analysis, decision to publish, or preparation of the manuscript.

### Grant Disclosures

The following grant information was disclosed by the authors:
Cystic Fibrosis Foundation: ZHANG12I0.
NIH: P20 RR017677.
DOD/DMRDP: DM090161.
South Carolina Clinical & Translational Research (SCTR) Institute: NIH/NCATS UL1 TR000062, NIH/NCATS TL1 TR000061.

### Competing Interests

The authors declare there are no competing interests.

### Author Contributions

- Jordon D. Gruber and Yong-Mei Zhang conceived and designed the experiments, performed the experiments, analyzed the data, wrote the paper, prepared figures and/or tables, reviewed drafts of the paper.
- Wei Chen performed the experiments, prepared figures and/or tables, reviewed drafts of the paper.
- Stuart Parnham performed the experiments, analyzed the data, prepared figures and/or tables, reviewed drafts of the paper.
- Kevin Beauchesne performed the experiments, reviewed drafts of the paper.
- Peter Moeller and Patrick A. Flume analyzed the data, reviewed drafts of the paper.

### Human Ethics

The following information was supplied relating to ethical approvals (i.e., approving body and any reference numbers):

This study was approved by the Institutional Review Board for Human Research of the Medical University of South Carolina: Pro00012798.

## Data Availability

The original data on all figures and DHQ levels in stable patient samples, and original GFP and phase-contrast images for Fig. 2, are supplied as Supplemental Information.

## Supplemental Information

Supplemental information for this article can be found online at http://dx.doi.org/10.7717/peerj.1495#supplemental-information.

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
