# Peer review of "The role of 2,4-dihydroxyquinoline (DHQ) in Pseudomonas aeruginosa pathogenicity"

_PeerJ, doi:10.7717/peerj.1495_

## Round 0.1 · original submission · Major Revisions

Both reviewers find this to be an interesting and , in places, innovative manuscript. The main finding, that DHQ is involved in P. aeuruginosa pathogenicity is a novel and significant finding. Reviewer 1 makes good points about the the lack of statistical information in a number of the figures. The number of replications, statistical tests used, degrees of freedom and p-vlaues need to be reported. It should be indicated in the legend whether error bars refer to SE, SD or con intervals. In addition, I advocate not using 'dynamite plots' (see link below) but use instead using strip plots (which show every data point in the experiment) with means and error bars overlayed.

http://biostat.mc.vanderbilt.edu/wiki/Main/DynamitePlots

Reviewer 1 ·

Basic reporting

The main findings reported in this manuscript are about the role of Pseudomonas aeruginosa quorum sensing quinolone signal DHQ in pathogenicity. The authors suggest that DHQ which is the product of pqs operon may be responsible for virulence. They also detected elevated levels of DHQ in the sputum of CF patients. Additionally, they provide evidence that DHQ can bind to PqsR and activate the transcription of pqs operon. These studies suggest an important role for quorum sensing molecule DHQ in P. aeruginosa pathogenicity.

The introduction contains relevant but still mostly general information, and presentation of this study is missing in an appropriate context. Motivation of new studies is not explained.

Experimental design

The procedure description in Materials and Methods is not accurate enough in some parts. For example, on p.6, it is not clear what kind of mutants and GFP-bacteria were used. What is the motivation of using C. elegans model? On p.7-8, the route of infection is not described. Statistical analysis is also missing.
Results and figures.
Fig. 1. What serve as a control here? How many experiments were performed? Please, provide indication of significant differences (*,**,***) in a graphs. Axis Y: Survival, (%).
Fig. 2. Quantification of colonization shown on fluorescence images collected from at least 3 experiments would be great to have in order to reach the conclusions about the differences in colonization.
Fig. 3, 4, 5, 8. Statistical analysis is not clear. What is presented here – means, SE or other? How many experiments were performed? Significant difference is not provided.
Based solely on the data in Fig. 1 and 2, I am not sure the authors can conclude that DHQ are the primary mediators of the increased virulence of P.aeruginosa. Therefore, I would recommend additional experiment on complementation of virulence of P. aeruginosa with synthetic DHQ.

Validity of the findings

Statistical analysis is not clear. The Discussion is somehow fragmented and does not interpret the results in light of previous knowledge.

Additional comments

No more comments.

Reviewer 2 ·

Basic reporting

The article meets the PeerJ standards

Experimental design

No issues with the experimental design. The methods employed are well described and reasonable for addressing the question of what role DHQ plays in Pseudomonas pathogenecity. The C. elegans model was a good fit for this study. Moreover, the use of HPLC for quantification of QS molecules in the culture supernatants and sputum was innovative.

Validity of the findings

Data reported are robust and support the conclusions of the article. Figures provide a strong representation of the different methodological angles taken to evaluate the role of DHQ in P. aeruginosa pathogenesis

Additional comments

Overall, this was an interesting study and the conclusions are based on layers of evidence. It provides more insight into the role of the Pseudomonas quinolone signal quorum sensing system in disease.

---

## Round 0.2 · accepted · Accept

You should consider making the raw data underlying the figures available either through additional supplemental files, or through a post-publication link to uploaded files on figshare or similar resource.